# Systematic Design of Crystal Structure for Hofmann-Like Spin Crossover Fe(L)$_2$[Ag(CN)$_2$]$_2$ Complexes

**Takashi Kosone** [1],*, **Yoshinori Makido** [2], **Syogo Okuda** [3], **Ayaka Haigo** [3], **Takeshi Kawasaki** [2], **Daisuke Akahoshi** [4], **Toshiaki Saito** [4] **and Takafumi Kitazawa** [2]

[1] Department of Science and Engineering, Graduate School of Science and Engineering, Tokyo Denki University, Hatoyama, Hiki-gun, Saitama 350-0394, Japan
[2] Department of Chemistry, Faculty of Science, Toho University, 2-2-1 Miyama, Funabashi, Chiba 274-8510, Japan
[3] Department of Creative Technology Engineering Course of Chemical Engineering, Anan College, 265 Aoki, Minobayashi, Anan, Tokushima 774-0017, Japan
[4] Department of Physics, Faculty of Science, Toho University, 2-2-1, Miyama 274-8510, Japan
* Correspondence: t-kosone@mail.dendai.ac.jp; Tel.: +81-49-296-2923

**Abstract:** The synthesis, crystal structures, and magnetic properties of a new two-dimensional (2D) Hofmann-like series, Fe$^{II}$(L)$_2$[Ag$^I$(CN)$_2$]$_2$ (L = 3-cyano-4-methylpyridine (**1**), allyl isonicotinate (**2**), phenyl-isonicotinate (**3**), and benzyl nicotinate (**4**)) were studied. These compounds have a 2D sheet structure because of their strongly determinate self-assembly process. An octahedral Fe$^{II}$ ion is coordinated with the nitrogen atoms of [Ag$^I$(CN)$_2$] linear units at equatorial positions and monodentate pyridine derivatives at the axial position. The layers construct a parallel stacking array. Compounds **1–3** show pairs of layers constructed by intermetallic Ag···Ag interactions. Compound **4** shows a mono-layer structure. The substituent bulk of the ligands affects the interlayer space. Compounds **1–4** undergo a 100% spin transition. However, compound **1**, incorporating a smaller group, has a relatively lower critical temperature ($T_c$ = 182 K (**1**), $T_c$ = 221 K (**2**), $T_c$ = 227 (**3**) and $T_c{}^1$ = 236 K, $T_c{}^2$ = 215 K (**4**)). We investigated the correlations between our systematic crystal design, substituent size, and the spin crossover profiles.

**Keywords:** coordination polymer; intermetallic interaction; cooperative interaction; crystal engineering

## 1. Introduction

Control of the self-assembly process enables the systematic design of supramolecular networks [1,2]. In this respect, the Hofmann-like coordination polymer [3] is a strong tool due to its well-defined two-dimensional (2D) sheet structure. Many types of Hofmann-like coordination polymer using various metal ions and pyridine derivatives ligands have been derived. Since the first Hofmann-like spin crossover (SCO) coordination polymer {Fe(py)$_2$[Ni(CN)$_4$]}$_n$ (py = pyridine) was reported [4], this structural motif has been frequently used to design Fe(II) SCO materials to enable us to determine the correlations between structural features and magnetic properties. The motif of the 2D structure consists of octahedral metal centers through N atoms of the cyanometalate unit at the equatorial position. However, some unexpected structures also occur in this process. Suppressing the structural diversity is required to systematically design the crystal structure and clarify the structure–property relationships. Since 2000, many 2D layers ({Fe$^{II}$(L)$_2$[M$^I$(CN)$_2$]$_2$}$_n$ [5–16], where M$^I$ = Ag or Au, L = monodentate pyridine derivatives) have been developed. These compounds exhibit a crystallographically similar structure. This structural consistency is much higher than that of the other Hofmann-like compounds.

Therefore, the consistency can be used to precisely modify the crystal structure. However, the applicable ligands for this bilayer system are still unknown, and only small bulky substituents have been investigated. So far, Hofmann-like 2D examples from the silver family, Fe(L)$_2$[Ag(CN)$_2$]$_2$, have been less studied than those of the gold series. In this study, we chose a variety of bulk sizes of the substituents, from less bulky (L = 3-cyano-4-methylpyridine (3-CN-4Mepy) (**1**)) to bulkier (L = allyl isonicotinate (Allyl-Isonic) (**2**), phenyl isonicotinate (Ph-Isonic) (**3**), benzyl nicotinate (Bz-Nic) (**4**)) substituents (Scheme 1), with the goal of demonstrating the systematic structural design and discussion of the SCO behavior.

**Scheme 1.** Molecular structure of the ligands of 3-cyano-4-methylpyridine, allyl isonicotinate, phenyl isonicotinate, and benzyl nicotinate.

## 2. Materials and Methods

### 2.1. Materials

All the chemicals were purchased from commercial sources (TOKYO CHEMICAL INDUSTRY CO., LTD, Chuo-ku, Japan) and used without any further purification.

### 2.2. Synthesis

#### 2.2.1. Preparation of Compound 1

Fleshly prepared single crystals of **1** were synthesized via the slow diffusion of two solutions, one of which contained a mixture of FeSO$_4$·(NH$_4$)$_2$SO$_4$·6H$_2$O (0.0980 g, $2.50 \times 10^{-4}$ mol) and 3-CN-4-Mepy (0.0591 g, $5.00 \times 10^{-4}$ mol) in 4 mL water. The other contained 8 mL ethanol-water (1:1) solution of K[Ag(CN)$_2$] (0.0996 g, $5.00 \times 10^{-4}$ mol). The two solutions were combined in a glass tube. Yellow single crystals suitable for single crystal X-ray diffraction were formed over 2 days. Due to the small amount of crystals produced and the mixed impurities in the glass tube, a powder sample for superconducting quantum interference device (SQUID), X-ray powder diffraction (XRPD), and elemental analysis was also prepared. One of these contained a mixture of FeSO$_4$·(NH$_4$)$_2$SO$_4$·6H$_2$O (0.1961 g, $5.00 \times 10^{-4}$ mol) and K[Ag(CN)$_2$] (0.0996 g, $5.00 \times 10^{-4}$ mol) in 20 mL ethanol-water (1:4). The other contained 8 mL water and 3-cyano-4-methylpyridine (0.0591g, $5.00 \times 10^{-4}$ mol). A yellow powder sample of **1** formed immediately. The powder sample was checked using XRPD data (Figure S1). Impurities and isomers were almost absent. Elem. Anal. Calcd for C$_{19}$H$_8$Ag$_2$FeN$_9$: C, 35.32; H, 1.98; N, 18.31. Found: C, 35.19; H, 2.10; N, 18.26. IR (cm$^{-1}$): 2229 ($\nu$CN (3-CN-4-mepy)), 2160 ($\nu$C≡N).

#### 2.2.2. Preparation of Compounds 2–4

Single crystals of complexes **2–4** were prepared following the same procedure as for **1**. The reaction mixture was allowed to stand undisturbed for at least 2 days. Single crystals grew slowly. For **4** only, large crystals were obtained and enough crystalline samples were observed using a binocular lens. The powder samples of **2** and **3** for SQUID, XRPD (Figure S1), and elemental analysis were prepared following the same procedure as **1**. The elemental analysis data satisfied the formula of the target compounds. Anal. Calcd for C$_{22}$H$_{18}$Ag$_2$FeO$_4$N (**2**): C, 37.63; H, 2.58; N, 11.97. Found: C, 37.46; H, 2.74;

N, 12.00. IR (cm$^{-1}$): 2163 ($\nu$C≡N). Calcd for C$_{28}$H$_{18}$Ag$_2$FeN$_6$O$_4$ (**3**): C, 43.45; H, 2.34; N, 10.86. Found: C, 43.22; H, 2.46; N, 10.61. IR (cm$^{-1}$): 2167 ($\nu$C≡N). Calcd for C$_{30}$H$_{22}$Ag$_2$Fe$_2$N$_6$O$_4$ (**4**): C, 44.92; H, 2.76; N, 10.48. Found: C, 44.91; H, 2.85; N, 10.62. IR (cm$^{-1}$): 2171 ($\nu$C≡N).

## *2.3. X-Ray Crystallography*

Data collection was performed on a BRUKER APEX SMART CCD area-detector diffractometer for **1–4** with Monochromed MoK$\alpha$ radiation ($\lambda$ = 0.71073 Å) (Bruker, Billerica, MA, USA). A selected single crystal was carefully mounted on a thin glass capillary and immediately placed under a liquid cooled N$_2$ stream. The diffraction data were treated using SMART and SAINT, and absorption correction was performed using SADABS [17]. The structures were solved using direct methods with SHELXTL [18]. All non-hydrogen atoms were refined anisotropically, and the hydrogen atoms were generated geometrically. Pertinent crystallographic parameters and selected metric parameters for **1–4** are displayed in Tables 1 and 2. Crystallographic data were deposited with Cambridge Crystallographic Data Centre (CCDC): Deposition numbers CCDC-1921426 for compound **1** (296 K), CCDC-1921428 for **1** (100 K), CCDC-1921429 for **2** (250 K), CCDC-1921430 for **2** (85 K), CCDC-1921431 for **3** (250 K), CCDC-1921432 for **3** (85 K), CCDC-1921433 for **4** (275 K), and CCDC-1921434 for **4** (90 K). These data can be obtained free of charge via http://www.ccdc.cam.ac.uk/conts/retrieving.html.

**Table 1.** Crystal data and structure refinement for compounds **1–4**.

| X | 1 (296 K) | 1 (100 K) | 2 (250 K) | 2 (85 K) |
|---|---|---|---|---|
| Empirical formula | C$_{18}$H$_{12}$Ag$_2$FeN$_8$ | C$_{18}$H$_{12}$Ag$_2$FeN$_8$ | C$_{22}$H$_{18}$Ag$_2$FeO$_4$N$_6$ | C$_{22}$H$_{18}$Ag$_2$FeO$_4$N$_6$ |
| Formula weight | 611.95 | 611.95 | 702.09 | 702.09 |
| Crystal size (mm$^3$) | 0.44 × 0.38 × 0.13 | 0.44 × 0.38 × 0.13 | 0.38 × 0.20 × 0.10 | 0.38 × 0.20 × 0.10 |
| Crystal system | Orthorhombic | Orthorhombic | Monoclinic | Monoclinic |
| $A$ (Å) | 14.7200 (7) | 14.1147 (10) | 12.2912 (6) | 12.4008 (19) |
| $B$ (Å) | 15.0169 (7) | 14.5991 (10) | 13.6582 (7) | 13.078 (2) |
| $C$ (Å) | 20.5702 (10) | 20.3274 (15) | 15.9764 (9) | 15.512 (2) |
| $V$ (Å$^3$) | 4547.0 (4) | 4188.7 (5) | 2677.0 (2) | 2515.7 (7) |
| $B$ (°) | | | 93.5210 (10) | 90.276 (2) |
| Space group | *Pbca* | *Pbca* | *P2$_1$/c* | *P2$_1$/c* |
| Z value | 8 | 8 | 4 | 4 |
| $D_{\text{calc}}$ | 1.788 | 1.941 | 1.742 | 1.857 |
| $F(000)$ | 2368 | 2368 | 1512 | 1376 |
| Reflections collected | 33,414 | 28,412 | 20,225 | 18,877 |
| Independent reflections | 7037 | 6174 | 7881 | 7605 |
| Parameters | 264 | 264 | 316 | 316 |
| Final $R_1$, $R_{\text{w}}$ ($I > 2s$) | 0.0336, 0.0777 | 0.0317, 0.0629 | 0.0326, 0.0624 | 0.0409, 1.118 |
| Final $R_1$, $R_{\text{w}}$ (all data) | 0.0622, 0.0910 | 0.0540, 0.0754 | 0.0738, 0.0710 | 0.0618, 1.333 |
| Goodness-of-fit | 1.080 | 1.143 | 0.805 | 0.819 |
| | 3 (250 K) | 3 (85 K) | 4 (275 K) | 4 (90 K) |
| Empirical formula | C$_{28}$H$_{18}$Ag$_2$FeN$_6$O$_4$ | C$_{28}$H$_{18}$Ag$_2$FeN$_6$O$_4$ | C$_{30}$H$_{22}$Ag$_2$Fe$_2$N$_6$O$_4$ | C$_{30}$H$_{22}$Ag$_2$Fe$_2$N$_6$O$_4$ |
| Formula weight | 774.07 | 774.07 | 802.13 | 802.13 |
| Crystal size (mm$^3$) | 0.24 × 0.23 × 0.20 | 0.24 × 0.23 × 0.20 | 0.28 × 0.24 × 0.22 | 0.28 × 0.24 × 0.22 |
| Crystal system | Monoclinic | Monoclinic | Monoclinic | Monoclinic |
| $A$ (Å) | 14.1172 (9) | 13.9673 (7) | 21.7109 (8) | 21.2491 (7) |
| $B$ (Å) | 13.6365 (8) | 13.1675 (7) | 10.6183 (4) | 10.2384 (3) |
| $C$ (Å) | 15.6922 (10) | 15.3684 (8) | 15.9222 (6) | 15.5439 (5) |
| $V$ (Å$^3$) | 3004.2 (3) | 2822.1 (3) | 3291.9 (2) | 3017.83 (16) |
| $B$ (°) | 96.0340 (11) | 93.1900 (10) | 116.2570 (10) | 116.82 |
| Space group | *P2$_1$/c* | *P2$_1$/c* | *C2/c* | *C2/c* |
| Z value | 4 | 4 | 4 | 4 |
| $D_{\text{calc}}$ | 1.712 | 1.902 | 1.618 | 1.765 |
| $F(000)$ | 1520 | 1608 | 1584 | 1584 |
| Reflections collected | 18,628 | 21,779 | 12,540 | 11,350 |

**Table 1.** *Cont.*

| Independent reflections | 6581 | 8529 | 4945 | 4517 |
|---|---|---|---|---|
| Parameters | 370 | 370 | 199 | 199 |
| Final $R_1$, $R_w$ ($I > 2s$) | 0.0329, 0.1028 | 0.0224, 0.0802 | 0.0262, 0.0415 | 0.0173, 0.0424 |
| Final $R_1$, $R_w$ (all data) | 0.0601, 0.1338 | 0.0299, 0.0935 | 0.0374, 0.0426 | 0.0186, 0.0429 |
| Goodness-of-fit | 0.813 | 0.702 | 1.776 | 1.091 |

$$R = (\Sigma ||Fo| - |Fc||)/\Sigma |Fo| \quad wR = \{\Sigma w(|Fo| - |Fc|)^2/\Sigma w|Fo|^2\}^{1/2}.$$

**Table 2.** Selected Fe-N bond lengths for **1–4**.

| 1 (296 K) | 1 (100 K) | 2 (250 K) | 2 (85 K) |
|---|---|---|---|
| Fe(1)–N(1): 2.162(3) | Fe(1)–N(1): 1.941(3) | Fe(1)–N(1): 2.218(2) | Fe(1)–N(1): 1.997(3) |
| Fe(1)–N(2): 2.129(3) | Fe(1)–N(2): 1.940(3) | Fe(1)–N(2): 2.237(2) | Fe(1)–N(2): 2.009(3) |
| Fe(1)–N(3): 2.144(2) | Fe(1)–N(3): 1.932(3) | Fe(1)–N(3): 2.148(2) | Fe(1)–N(3): 1.938(3) |
| Fe(1)–N(4): 2.136(2) | Fe(1)–N(4): 1.944(3) | Fe(1)–N(4): 2.140(2) | Fe(1)–N(4): 1.941(3) |
| Fe(1)–N(5): 2.234(3) | Fe(1)–N(5): 2.003(3) | Fe(1)–N(5): 2.149(2) | Fe(1)–N(5): 1.939(3) |
| Fe(1)–N(6): 2.231(3) | Fe(1)–N(6): 1.994(3) | Fe(1)–N(6): 2.118(3) | Fe(1)–N(6): 1.937(3) |
| **3 (250 K)** | **3 (85 K)** | **4 (275 K)** | **4 (90 K)** |
| Fe(1)–N(1): 2.198(3) | Fe(1)–N(1): 2.017(2) | Fe(1)–N(1): 2.149(1) | Fe(1)–N(1): 2.003(1) |
| Fe(1)–N(2): 2.198(3) | Fe(1)–N(2): 1.995(2) | Fe(1)–N(2): 2.149(1) | Fe(1)–N(2): 1.926(2) |
| Fe(1)–N(3): 2.102(3) | Fe(1)–N(3): 1.934(2) | Fe(1)–N(3): 2.149(2) | Fe(1)–N(3): 1.934(1) |
| Fe(1)–N(4): 2.099(3) | Fe(1)–N(4): 1.932(2) | Fe(1)–N(4): 2.141(2) | Fe(1)–N(4): 1.914(1) |
| Fe(1)–N(5): 2.101(3) | Fe(1)–N(5): 1.929(2) | | |
| Fe(1)–N(6): 2.107(3) | Fe(1)–N(6): 1.934(2) | | |

## 2.4. Magnetic Measurements

We measured the temperature dependence of the magnetic susceptibility of complexes **1–4** of the powdered samples in the temperature range of 2–300 K with a cooling and heating rate of 2 K·min$^{-1}$ in a 1 kOe field on an Quantum Design MPMS-XL SQUID magnetometer (Quantum Design, Inc., Pacific Center Court San Diego, CA, USA). The diamagnetism of the samples and sample holders were considered.

## 3. Results

### 3.1. Crystal Structures

#### 3.1.1. Overview of Bilayer Structure for 1–3

The constructed frameworks for **1–3** are almost isostructural. However, in terms of crystal system, compound **1** is orthorhombic, whereas **2** and **3** are monoclinic. The asymmetric units of **1–3** consist of an octahedral Fe$^{II}$ ion coordinated with the nitrogen atoms from [Ag$^I$(CN)$_2$] linear units at equatorial positions and monodentate py ligands at axial positions (Figure 1). [Ag$^I$(CN)$_2$] units produce an rectangular mesh layer woven by –Ag–N–C–Fe–C–N–Ag– infinite chains (Figure 2). A gap inside the rectangular is penetrated by py ligands from the upper and lower layers (Figure 2). The adjacent two layers interact via argentophilic interactions, which form a bilayer structure (Figure 3).

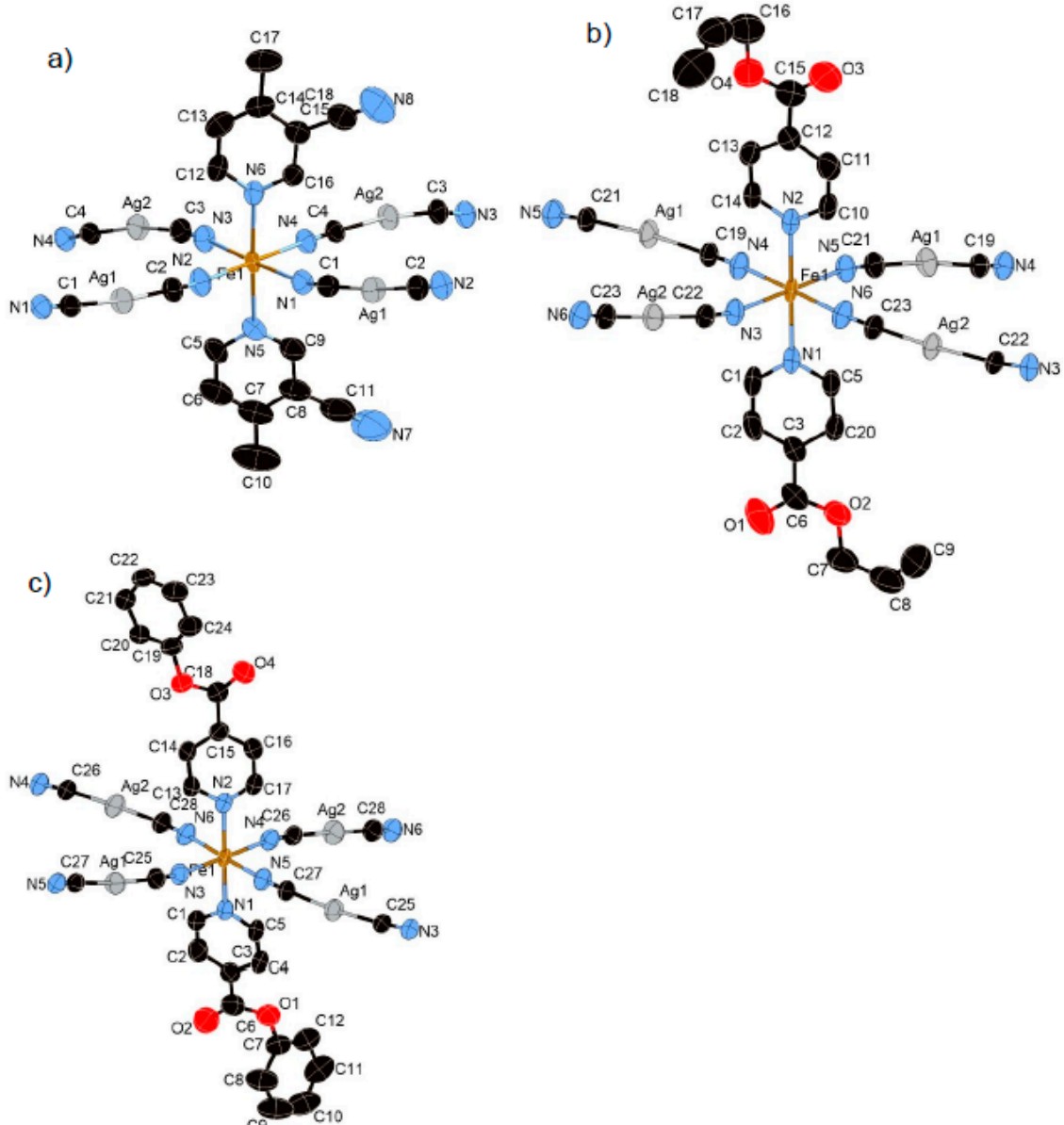

**Figure 1.** Coordination structures of (**a**) **1**, (**b**) **2**, and (**c**) **3** containing its asymmetric unit (HS state). In these pictures, the hydrogen atoms are omitted for clarity.

### 3.1.2. Structure of Compound 1 (T = 296 K and 100 K)

Compound **1** crystallized in the orthorhombic centrosymmetric space group *Pbca*. The relative position of the substituents of two 3-CN-4-Mepy ligands in [$Fe^{II}$(3-CN-4-Mepy)$_2$][$Ag^I$(CN)$_2$]$_2$ asymmetric unit are in *cissoid* conformation (Figure 1a). The Fe–N$_{py}$ bond lengths (average distance of Fe–N$_{py}$ = 2.233(4) Å (296 K), 1.999(4) Å (100 K)) and Fe–N$_{CN}$ bond lengths (average distance of Fe–N$_{CN}$ = 2.143(5) Å (296 K), 1.939(6) Å (100 K)) correspond to the usual values for $Fe^{II}$ 100% HS state and 100% LS state. The closest Ag(1)···Ag(2) distance in the bilayer is 3.0085(4) (296 K) and 2.9489(4) (100 K) Å, less than the sum of the van der Waals radii of Ag.

### 3.1.3. Structure of Compound 2 (T = 250 K and 85 K)

The bilayer structure of **2** is almost similar to that of **1**. However, the lattice constant is much higher than that of **1** and the distance between adjacent bilayers is larger. The Fe–N$_{py}$ bond lengths (average distance of Fe–N$_{py}$ = 2.228(3) Å (250 K), 2.003(4) Å (85 K)) and Fe–N$_{CN}$ bond lengths (average

distance of Fe–N$_{CN}$ = 2.139(5) Å (250 K), 1.939(4) Å (85 K)) correspond well to the usual values for Fe$^{II}$ 100% HS state and 100% LS state. The closest Ag(1)···Ag(2) distance in the bilayer is 3.289(1) (250 K) and 3.340(2) (85 K) Å, less than the sum of the van der Waals radii of Ag. Notably, there is an additional intra-bilayer interaction between the coordinatively unsaturated Ag(I) center and the oxygen atom of the oxo group (closest Ag(1)···O(1) distance = 2.905(3) (250 K) and 2.934(3) (85 K) Å), which is smaller than the sum of the van der Waals radii. Therefore, the Ag ion is defined as a pseudo trigonal coordination geometry (bond angle for C(23)–Ag(1) –O(1) = 81.433°(250 K) and 85.490°(85 K)) (Figure S2a). As a result, the Ag···O interaction strongly enhances the three-dimensional (3D) networks between consecutive bilayers (Figure 3b).

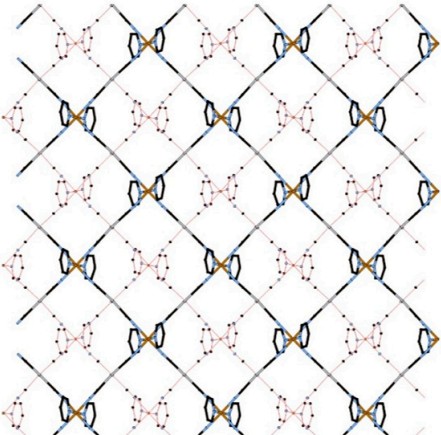

**Figure 2.** View of the bilayer of **1** (HS state) along the *c* axis (lower layer: red line, upper layer: cylinders). In this picture, the substituents of the ligands are omitted for clarity.

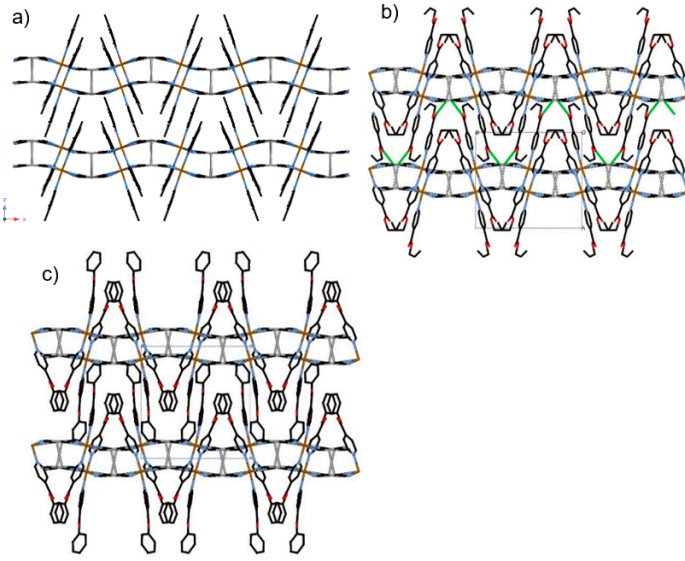

**Figure 3.** View of stacking layers of (**a**) **1**, (**b**) **2**, and (**c**) **3** in HS state involved in Ag··Ag interactions, as indicated by grey lines, and Ag···O interactions, as indicated by green lines.

### 3.1.4. Structure of Compound 3 (T = 250 K and 85 K)

The lattice constant for **3** is even higher. The distance between adjacent bilayers is the largest. The Fe–N$_{py}$ bond lengths (average distance of Fe–N$_{py}$ = 2.198(3) Å (250 K), 2.006(2) Å (85 K)) and Fe–N$_{CN}$ bond lengths (average distance of Fe–N$_{CN}$ = 2.102(3) Å (250 K), 1.925(2) Å (85 K)) correspond well to the usual values for Fe$^{II}$ 100% HS state and 100% LS state. The closest Ag···Ag distance in the bilayer is 3.410(1) (250 K) and 3.416(1) (85 K) Å. Despite the similar isonicotinic substituents, in contrast

to **2**, there are no significant Ag···O contacts between adjacent bilayers (Ag(2)···O(4) = 4.143(4) Å (250 K)). Apparently, the interlayer space is larger than that of **2**.

### 3.1.5. Structure of Compound 4 (T = 275 K and 90 K)

Complex **4** crystallizes in the centrosymmetric monoclinic space group *C2/c*. The complex consists of the bimetallic $Fe^{II}Ag^{I}$ unit of $Fe(Bz\text{-}nic)_2[Ag^{I}(CN)_2]_2$ (Figure 4a). For this complex only, the Fe(1) atom lies on an inversion center. The $Fe–N_{py}$ bond lengths (average distance of $Fe–N_{py}$ = 2.149(1) Å (275 K), 2.003(1) Å (90 K)) and $Fe–N_{CN}$ bond lengths (average distance of $Fe–N_{CN}$ = 2.146(2) Å (275 K), 1.932(6) Å (90 K)) correspond well to the usual values for $Fe^{II}$ 100% HS state and 100% LS state. The crystal structure of **4** is not a bilayer structure but a flat mono-layer structure woven by straight chains (bond angles for Fe(1)–N(2)–C(14) and Fe(1)–N(3)–C(15)= 180.0°) and waved chains (bond angle for Fe(1)–N(2)–C(14) = 163.54°) (Figure 4b). In contrast with **1**–**3**, the distorted rectangular window is not penetrated by the ligand. Therefore, the bulk induces an expansion of the interlayer space, which breaks the Ag–Ag interaction (Ag···Ag = 8.129(2) Å (275 K) and 7.9596(2) Å (90 K)). However, the Ag···O interaction, which is weaker than that of **2**, remains in this structure (Ag(1)···O(2) = 3.192(2) Å (27 K) and 3.079(1) Å (90 K)), which connects each layer (Figure 4c). Thus, the Ag ion has a pseudo square-planar coordination geometry (bond angle for C(14)–Ag(1)–O(2) = 78.35°) (Figure S2b).

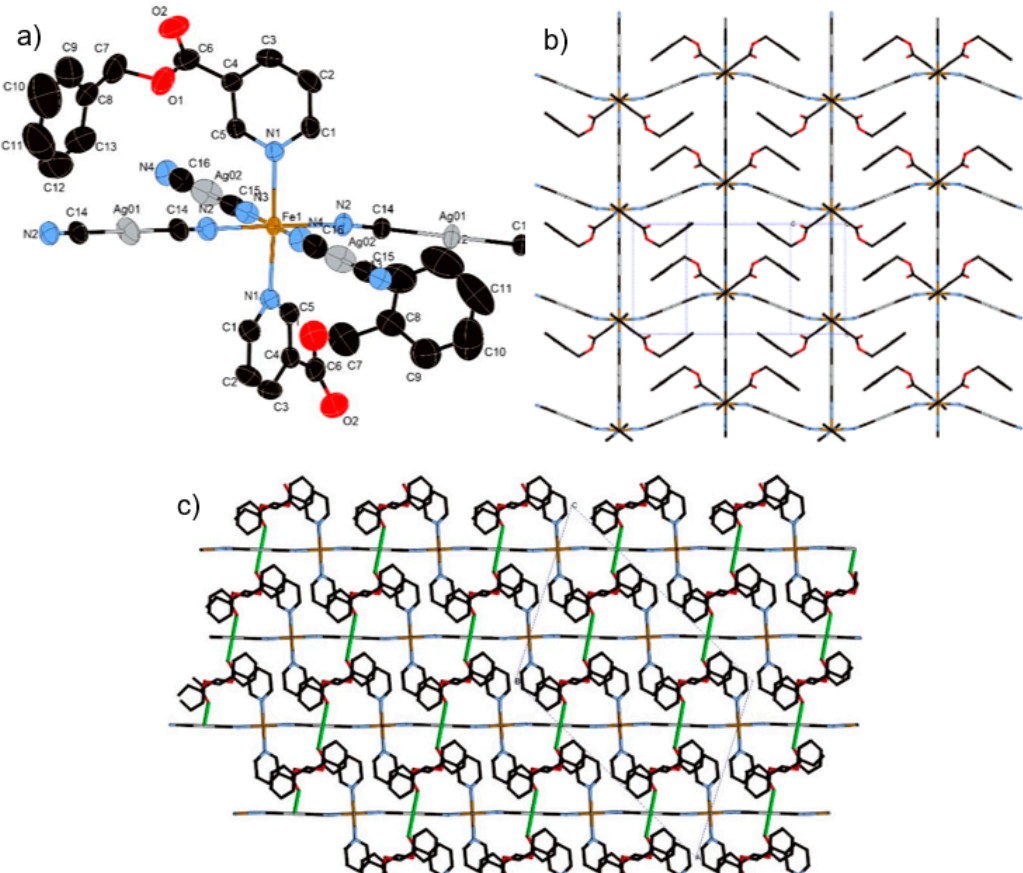

**Figure 4.** (**a**) Coordination structure of **4** containing its asymmetric unit, (**b**) view of the two-dimensional (2D) layer of **4**, and (**c**) stacking layers of **4** involved in Ag···O interactions as indicated by green lines.

### 3.2. Magnetic Properties

#### 3.2.1. Thermal Dependence Magnetic Behavior of Compound 1

Figure 5 shows the thermal dependence of $\chi_M T$ for **1**. At room temperature, $\chi_M T$ was 3.75 cm³·K·mol⁻¹. Upon cooling, $\chi_M T$ remains almost constant down to 230 K; below this temperature, $\chi_M T$ shows a 100% spin conversion. ($T_c$ = 182 K).

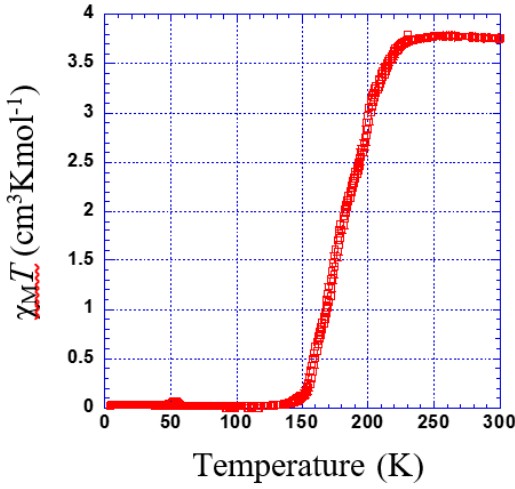

**Figure 5.** Thermal dependence of $\chi_M T$ plot for **1**.

#### 3.2.2. Thermal Dependence Magnetic Behavior of Compound 2

$\chi_M T$ versus $T$ plotted for **2** is shown in Figure 6. The $\chi_M T$ value is constant in the range of 230 to 300 K. Just below 230 K, a sharp transition is observed ($T_c$ = 221 K). However, there might be a very slight step separated by a narrow plateau (see inset of Figure 6).

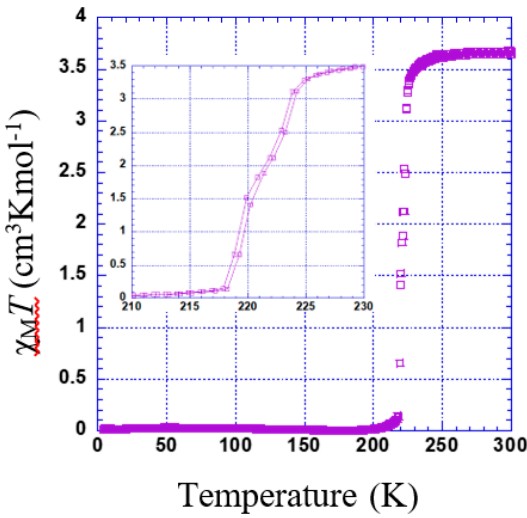

**Figure 6.** Thermal dependence of $\chi_M T$ plot for **2**. Inset shows the range of 210 to 230 K.

#### 3.2.3. Thermal Dependence Magnetic Behavior of Compound 3

$\chi_M T$ versus $T$ plotted for **3** is shown in Figure 7. $\chi_M T$ was 3.17 cm³·K·mol⁻¹. Upon cooling, $\chi_M T$ remains almost constant down to 260 K; below this temperature, $\chi_M T$ shows a gradual decrease to 0.6 cm³ K mol⁻¹ at 220 K ($T_c$ = 227 K) indicating an incomplete spin transition. The second decrease in the residual value of $\chi_M T$ at the lower temperature is due to the typical behavior of zero-field splitting (ZFS).

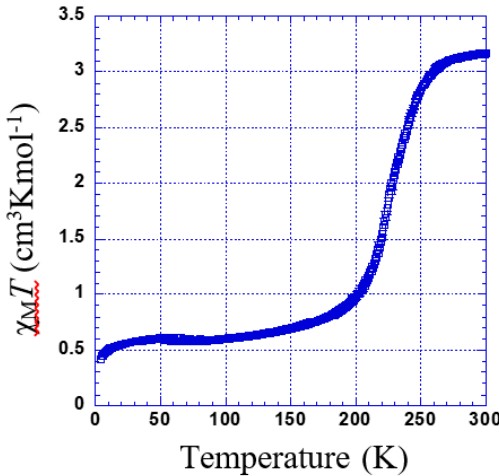

**Figure 7.** Thermal dependence of $\chi_M T$ plot for **3**.

### 3.2.4. Thermal Dependence Magnetic Behavior of Compound 4

$\chi_M T$ versus $T$ plotted for **4** is shown in Figure 8. The $\chi_M T$ is constant within 260–300 K. Below this temperature, $\chi_M T$ shows two marked steps. In the first step, a gradual decrease to 1.63 cm$^3$ K mol$^{-1}$ at 220 K indicates an almost 50% spin transition ($T_c{}^1$ = 236 K). Then, it falls rapidly to 0.1 cm$^3$ K mol$^{-1}$ ($T_c{}^2$ = 215 K).

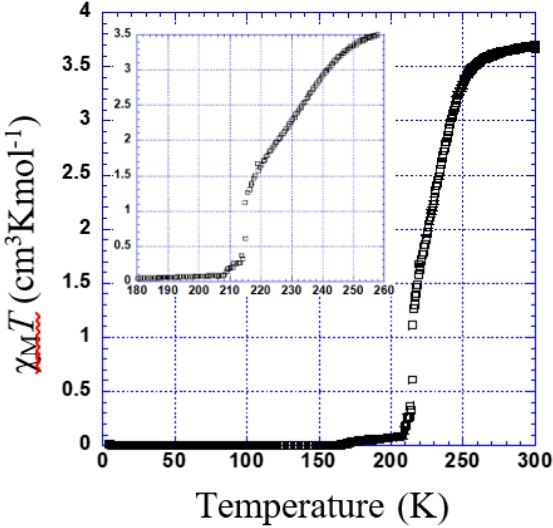

**Figure 8.** Thermal dependence of $\chi_M T$ plot for **4**. Inset shows the temperature range of 180 to 260 K.

## 4. Discussion

Initially, we discussed the series of bilayer structures, except for the mono-layer of **4**. Other members of similar Fe–Ag bilayer structures of {Fe(X-py)$_2$[Ag(CN)$_2$]$_2$} (X = 3-F, 3-Cl, 3-Br) have been reported [6]. These analogous compounds were also discussed and compared to discover the correct means of crystal designing Hofmann-like structures.

Cell volumes, SCO behavior, and $T_c$ for **1–3** and previous compounds are listed in Table 3. The cell volume expanded considerably with increasing substituent bulk. Despite the lattice expansion, these compounds completely maintain their bilayer structure. In a previous paper, we investigated the Fe–Au analogous series [16], which shows the same cell expansion tendency. In the present, we demonstrated the wider range of substituent sizes available to systematically construct the bilayer structure.

**Table 3.** Cell volumes, SCO behavior, and critical temperature ($T_c$) of the bilayer series for {$Fe^{II}(X\text{-py})_2[Ag^I(CN)_2]_2$} in HS state.

| Substituents (X) | Cell Volume ($Å^3$) | SCO Behavior Type | Critical Temperature $T_c$ (K) | Reference |
|---|---|---|---|---|
| 3-F | 1934.1 | gradual (2-step) | 162(1st), 96 (2nd) | [6] |
| 3-Cl | 1959.7 | steep (1-step) | 106 | [6] |
| 3-Br | 1959.4 | none | None | [6] |
| 3-CN-4-Me (**1**) | 4547.0 * | gradual (1-step) | 182 | - |
| Allyl-Isonic (**2**) | 2677.0 | steep (1-step) | 221 | - |
| Ph-Isonic (**3**) | 3004.2 | gradual (1-step) | 227 | - |

* The unit cell contains two bilayers, so the cell volume is twice as large as the other unit cell of this series.

Despite the similar large bulk of Ph-Isonic, the Bz-Nic ligand produces a different 2D stacking layer structure without strong Ag–Ag interactions. The reason for the structural difference is likely the direction of the bulk. In terms of the substituents' positions, the substituents exist at the 3-position and 4-position of the Bz-Nic and Ph-Isonic ligand, respectively. For the Ph-Isonic ligand, the substituent molecule is well-folded and oriented vertical to the layer (Scheme 2a). Therefore, the width of the substituent enables easy access to the rectangle window $Fe_4[Ag^I(CN)_2]_4$ (Scheme 2b). In contrast to Ph-Isonic, the orientation of the Bz-Nic substituent is almost parallel to the layer, which has a larger bulk compared to the window space (Scheme 2c). Even if the rotation of the Fe–$N_{py}$ bond axis is free, a portion of the ligand collides with the edge of the window. In other words, the monolayer structure of **4** has a lower potential energy than the other possible supramolecular isomers. One additional example of a flat monolayer Fe[4-(3-pentyl)pyridine]$_2$[Au$^I$(CN)$_2$]$_2$·(guest) (guest = 4-(3-pentyl)pyridine) [19]. 4-(3-pentyl) substituent is also along the 2D sheet, which blocks the penetration of the window.

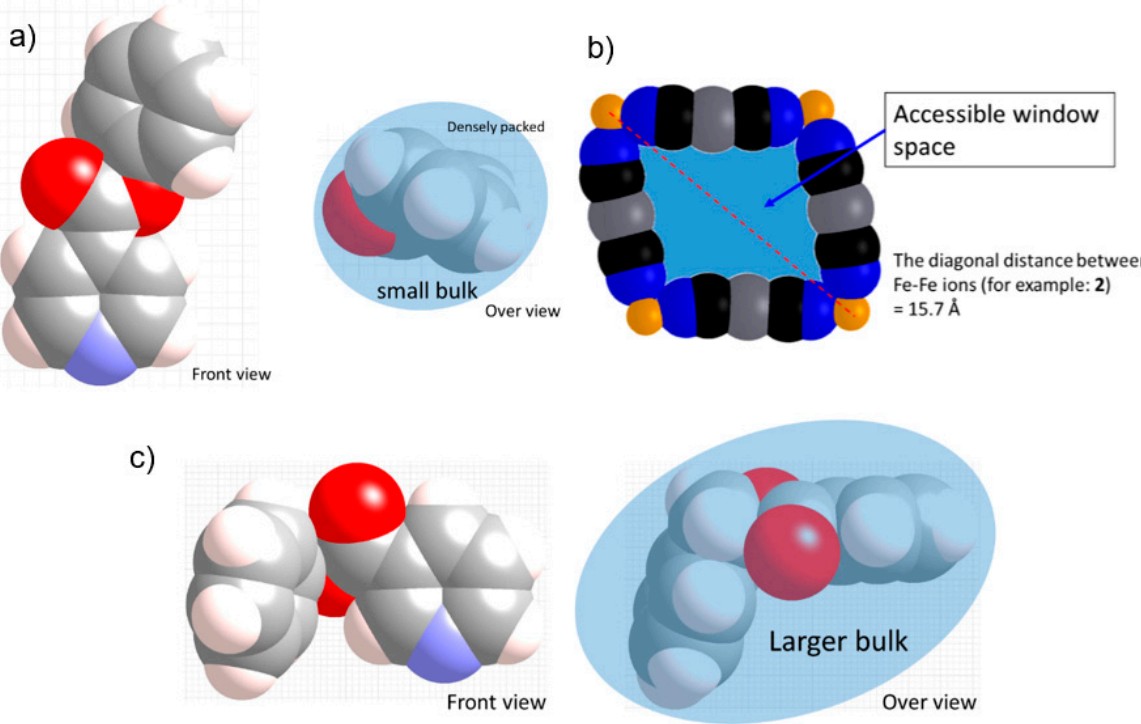

**Scheme 2.** Representation of the substituent bulk of the (**a**) Allyl-Isonic and (**c**) Ph-Isonic compounds. (**b**) Representation of the rectangle window. These pictures were drawn using the CPK model of the crystal structure data.

The previously reported $\{Fe^{II}(L)_2[Ag^I(CN)_2]_2\}_n$ series and compound **1**, which has small bulky substituents, show relatively lower $T_c$ ($T_c$ = 90–180 K) compared with the large bulky substituent groups for **2** and **3** ($T_c$ = 221 K (**2**), 227 K (**3**)). Although both substituents are electron-withdrawing according to the Hammett equation [20], the large bulk effects seem to increase the $T_c$. This higher $T_c$ could be attributed to the steric effect that produces chemical pressure on the $Fe^{II}$ centers. The $T_c$ order aligns with the expansion of the lattice volume. Considering the steric effect hypothesis, **3** is the most effective. The average Fe–N bond length of **3** is much shorter than that of the other compounds (3-F: 2.189 Å, 3-Cl: 2.183 Å, 3-Br: 2.185 Å, 3-CN-4Me: 2.189 Å, Allyl-Isonic: 2.173 Å, Ph-Isonic: 2.134 Å). However, the $T_c$ for **2** and **3** are almost same. The reason for this could be the competition with the electron withdrawing, in which the phenyl group is stronger than that of the allyl group.

The magnetic behavior of **2** shows the most rapid spin transition, which explain the strongest cooperativity derived from the Ag–Ag intermetallic interaction and unique and strong Ag–O interaction. These extensive cooperative networks affect the SCO profiles.

## 5. Conclusions

New 2D supramolecular networks $[Fe^{II}(L)_2[Ag^I(CN)_2]_2]$ exhibiting spin transitions were reported in this paper. In this study, we identified additional applicable ligands for constructing bilayer structures. This template structure strongly supports the self-assembly process, which would enable the precise design of crystal structures and physical properties. This systematic structural change for the Hofmann-type spin crossover family is useful for investigating and modifying the spin crossover phenomena.

**Supplementary Materials:** The following are available online at http://www.mdpi.com/2073-4352/9/7/370/s1, Figure S1: X ray Powder diffraction (red line) and simulation powder pattern (black line) for **1** (a), **2** (b), **3** (c) and **4** (d); Figure S2: Coordination environment of Ag(I) ion for **2** (a) and **4** (b)

**Author Contributions:** Data curation, Y.M., S.O., and A.H.; Formal analysis, T.Ka., D.A., and T.S.; Investigation, T.Ko., Y.M., S.O., and A.H.; Project administration, T.Ko.; Supervision, T.Ki.

**Funding:** This work was financially supported by KAKENHI (JSPS/15K05485 and 18K04964) and the Yashima Environment Technology Foundation. Part of this work was supported by the Ministry of Education, Culture, Sports, Science, and Technology, Japan (MEXT)-Supported program for the Strategic Research Foundation at Private Universities 2012–2016.

**Conflicts of Interest:** The authors declare no conflict of interest.

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
