# Peer review of "Systematic Design of Crystal Structure for Hofmann-Like Spin Crossover Fe(L)2[Ag(CN)2]2 Complexes"

_crystals, doi:10.3390/cryst9070370_

Round 1
Reviewer 1 Report
This paper described a crystal structure of supramolecular networks with [FeII(L)2[AgI(CN)2]2] and their spin-crossover behavior. In this research, the authors controlled the layered structure of [FeII(L)2[AgI(CN)2]2] by substitution of ligand molecule, L. The bulkiness of substituent ligand gave the shift of spin-crossover temperature.
The aim of these experiments and the explanation of their results are acceptable. The manuscript is worth publishing in Crystals after minor revisions. Please consider following subjects.
line 123
"The closet Ag...Ag distance in the bilayer"
Where is the closest Ag...Ag contact? Is it gray lines in figures 3? Gray lines in figure 3 seems like a "bond" between Ag...Ag, instead of the van der Waals contact.
Could you define the place of them in the figure 3?
line 169-170
"χMT shows 100% spin conversion"
Is there any explanation on the hysteresis loop in the temperature dependence of χMT for sample 2? Are there no hysteresis loops in the temperature dependence of χMT for sample 1, 3and 4? What is the difference between existence and absence of hysteresis loop in χMT plots?
line 178-179
"The second decrease in the value of χMT at lower temperature is due to a typical behavior of zero-field splitting (ZFS)."
Why only sample 3 shows the ZFS? Is there anisotropic coordination environment only for sample 3? Please explain the structural feature which affect to the ZFS for sample 3.
*Are there some errors in writing?
line 73
“C19H8Au2FeN9” -> “C19H8Ag2FeN9”?
line 116
"argentphilic" -> " argentophilic"
line 120
"cisoid" -> "cissoid"?
line 123, 134, 137, 146
"closet" -> " closest"?
line 169, 176
"m3 K mol-1" -> "c m3 K mol-1"
line 200, 201, 203
"Scheme 3a" etc. -> "scheme 2a" etc.?
There is disunity in notation of apparatuses.
line 105 "MPMS-XL Quantum Design SQUID magnetometer"
line 88 "BRUKER APEX SMART CCD area-detector diffractometer"
Usually, it is suitable in order of "maker, model number, usage of measurement devices", I think. Please unify them.
Author Response
Thank you for your comments. I send a revised file.
We have corrected according to your comments. More explanation to the comments are described below.
>line 123
>"The closet Ag...Ag distance in the bilayer"
>Where is the closest Ag...Ag contact? Is it gray lines in figures 3? Gray lines in figure 3 seems
> like a "bond" between Ag...Ag, instead of the van der Waals contact.
>Could you define the place of them in the figure 3?
We have corrected the figure caption.
>line 169-170
>"χMT shows 100% spin conversion"
>Is there any explanation on the hysteresis loop in the temperature dependence of χMT for sample 2? Are
> there no hysteresis loops in the temperature dependence of χMT for sample 1, 3and 4? What is the
>difference between existence and absence of hysteresis loop in χMT plots?
Actually, there are subtle difference between sample2 and other samples. We decided the existence or absence of hysteresis loop by only the apperance of the behavior of the cooling and heating process.
>line 178-179
>"The second decrease in the value of χMT at lower temperature is due to a typical behavior of
>zero-field splitting (ZFS)."
>Why only sample 3 shows the ZFS? Is there anisotropic coordination environment only for
>sample 3? Please explain the structural feature which affect to the ZFS for sample 3.
Samples 1, 2 and 4 are 100% spin transition. So There are no more decreasing observed at lower temperature.
Reviewer 2 Report
This manuscript deals with new Hofmann-like spin-crossover complexes consisting of a cyanometallate coordination network. Crystal structures of the compounds in both the high- and low-temperature phases were determined by means of single-crystal X-ray diffraction analyses, and spin-transition behavior was investigated by using a SQUID magnetometer. All the samples show thermally-induced spin transition. The authors discussed the relationship among the bulkiness of the pyridine ligand, crystal packing, and transition temperature of spin-crossover behavior. Such a result and discussion is informative and interesting for an audience in Crystals. Unfortunately, the manuscript writing is somewhat awkward. In addition, there are a lot of misdescriptions some of which are scientifically serious errors. However, this study is clear and relatively important in the viewpoint of magneto-structure correlation. The paper submitted might be acceptable after major revision.
(1) Section 2.2.1. Please check molar quantities of the reagents.
(2) Line 113. [Ag(CN)2] is not bidentate. Please revise the sentence.
(3) Line 143. “The lattice constant for 3 is further higher.” is not appropriate because the cell volume of complex 3 is smaller than those of complexes 1 and 4. The authors should rephrase it with a more clear expression.
(4) Line 153. For complex 4, Fe(Bz-nic)2[Ag(CN)2]2 is not crystallographically an asymmetric unit because the complex has one crystallographically unique Bz-nic ligand. I recommend this sentence should be altered.
(5) Line 172. “The chiT value is constant in the range 215-300 K” is not appropriate because of the confliction with the following explanation: “Just below 230 K, …”.
(6) Line 179. Why is the decrement of chiT due to zero-field splitting? At low temperatures, the iron ion shows the low-spin state (S = 0), which is diamagnetic. Some residual chiT value were observed at low temperatures, suggesting a magnetic impurity or incomplete transition nature. This problem needs to be addressed.
(7) On scientific terms, I recommend the following word should be altered: (a) bulk effect à size effect or steric effect, (b) van der Waals radius model à CPK model or space-filling model
[Spelling variants]
(1) British and American spellings are mixed in this manuscript (behaviour/behavior, centre/center).
(2) 2-D and 2D are used together.
[Spelling errors and others]
In places. closet à closest
Line 37. have à has
Line 37. heavily à frequently
Line39. consist à consists
Line 40. cyano metalate à cyanometalate (or cyanidometalate)
Line 105. a MPMS-XL à an MPMS-XL
Line 116. argentphilic à argentophilic
Line 136. coordinative unsaturated à coordinatively unsaturated
Line 138. radius à radii
Line 138. Ag ion defines à Ag ion is defined as
Line 163. layers à layer
Lines 169 & 176. m3 K mol-1 à cm3 K mol-1
Line 187. [Au(CN)2]2 à [Ag(CN)2]2
Line 187. collide to à collides with
Author Response
Thank you for your comments. I send a revised file.
We have checked and corrected according to your comments. More explanation to the comments are described below.
>(6) Line 179. Why is the decrement of chiT due to zero-field splitting? At low temperatures, the
>iron ion shows the low-spin state (S = 0), which is diamagnetic. Some residual chiT value were
>observed at low temperatures, suggesting a magnetic impurity or incomplete transition nature.
>This problem needs to be addressed.
Actually, There must be this is incomplete spintransition. So we added the sentence. Maybe there is no impurities because of the results of the XRPD and elemental analysis.
Reviewer 3 Report
The paper submitted for publication in “Crystals” by Pr Kosone and collaborators is entitled “ Systematic Designing of Crystal Structure for Hofmann-like Spin Crossover Complexes [Fe(L)2[Ag(CN)2]2.” This work aims to control of self-assembly process in order to a systematic designing of supramolecular network.
Four new systems, based on functionalized pyridine ligands such as cyano methyl pyridine, allylisonicotinate, phenylisonicotinante and benzylnicotinate, have been prepared and fully characterized by X-ray diffraction. Their magnetic properties are discussed as well.
This paper is almost clearly presented but would deserve to be enhanced owing to some inaccuracies, which are listed below.
First, complex 2 crystallises in the monoclinic P 21/c space group at rt, but the authors ascertained that it is still in this monoclinic space group at low temperature (85 K) even though the b angle value is 90 °. In the chek cif file this point is mentioned as a alert level B.
This point is not really discussed in this paper. The comment line 127 is not sufficient at all. “…the determined space group and lattice parameter are a little strange…”.
Did the authors try to solve the structure in an orthorhombic group? And if yes, why they did not discussed this? Is it possible in this case to envision a structural phase transition during the thermal variation?
Line 72, The expression “the two solutions were communicated” is not clear, What did the authors want to say?
Line 225 The sentence “It is important that the selection of the substituent size and the orientation. 3 is not clear.
Besides I have noticed few mistakes
Line 134 and others. The Closet Ag Ag distance, did the authors cite the closest Ag Ag distance ?
Line 150 the distance between Ag and O also becomes longer.
Line 210 although the both substituents
Author Response
Thank you for your comments. I send a revised file.
We have corrected according to your comments. More explanation to the comments are described below.
>First, complex 2 crystallises in the monoclinic P 21/c space group at rt, but the authors
>ascertained that it is still in this monoclinic space group at low temperature (85 K) even though the
>b angle value is 90 °. In the chek cif file this point is mentioned as a alert level B.
We have tried to solve the structure in an monoclinic group with the beta angle
90.276 degree. This structure was well solved. We deposit the structure again.
Reviewer 4 Report
Kosone et al reported systematic designing of Hofmann-type Spin Crossover Complexes Fe(L)2[Ag(CN)2]2. The results are very interesting to the field of SCO materials. The compounds are characterized acceptably to a minimum standard, but there are no additional measurements to explain how easily the solvent is lost from the samples, and how much that affects their magnetic behaviour. TGA data would be useful in that respect. In addition, English should be improved (see below a few examples).
Therefore, my recommendation is acceptance after minor revision.
More references about Hofmann-type SCO should be added.
Line 21, “at axial position should be added after “pyridine derivatives”
Line 24 “which has smaller bulk” may be replaced by “incorporating a smaller group”
Line 34 I don’t understand why “polymer is a strong tool”.
Line 43, “from the 2000’s up to now” may be replaced with “Since 2000,”
There are other errors in English. Please carefully check the article!
Author Response
Thank you for your comments. I send a revised file.
We have corrected according to your comments. More explanation to the comments are described below.
>there are no additional measurements to explain how easily the solvent is lost from the samples,
>and how much that affects their magnetic behaviour. TGA data would be useful in that respect.
Our compounds have no solvent. So We didn't carried out TG/DTA measurement.
>More references about Hofmann-type SCO should be added.
We have added.
Round 2
Reviewer 3 Report
Given the appropriate corrections, the manuscript submitted for publication in “Crystals” by Pr Kosone and collaborators deserves to be published after minor revision.
In my opinion there are still some uncorrected mistakes but but essentially typographical.
Such as despite instead of despite and others.
Otherwise, the scientific presentation is now right. Compound 2 is crystallises effectively in the monoclinic space group.
Author Response
Thank you for your comments.
Our typographical errors are gonig to be corrected by native speakers.